# Submassive Pulmonary Embolism: Current Perspectives and Future Directions

**DOI:** 10.3390/jcm10153383

**Published:** 2021-07-30

**Authors:** Phillip C. Nguyen, Hannah Stevens, Karlheinz Peter, James D. McFadyen

**Affiliations:** 1Department of Haematology, Alfred Hospital, Melbourne, VIC 3181, Australia; p.nguyen@alfred.org.au (P.C.N.); hannah.stevens@monash.edu (H.S.); 2Atherothrombosis and Vascular Biology, Baker Heart and Diabetes Institute, Melbourne, VIC 3004, Australia; karlheinz.peter@baker.edu.au; 3Department of Medicine, Central Clinical School, Monash University, Melbourne, VIC 3800, Australia; 4Baker Department of Cardiometabolic Health, University of Melbourne, Melbourne, VIC 3010, Australia; 5Department of Cardiology, The Alfred Hospital, Melbourne, VIC 3181, Australia

**Keywords:** pulmonary embolism, submassive

## Abstract

Submassive pulmonary embolism (PE) lies on a spectrum of disease severity between standard and high-risk disease. By definition, patients with submassive PE have a worse outcome than the majority of those with standard-risk PE, who are hemodynamically stable and lack imaging or laboratory features of cardiac dysfunction. Systemic thrombolytic therapy has been proven to reduce mortality in patients with high-risk disease; however, its use in submassive PE has not demonstrated a clear benefit, with haemodynamic improvements being offset by excess bleeding. Furthermore, meta-analyses have been confusing, with conflicting results on overall survival and net gain. As such, significant interest remains in optimising thrombolysis, with recent efforts in catheter-based delivery as well as upcoming studies on reduced systemic dosing. Recently, long-term cardiorespiratory limitations following submassive PE have been described, termed post-PE syndrome. Studies on the ability of thrombolytic therapy to prevent this condition also present conflicting evidence. In this review, we aim to clarify the current evidence with respect to submassive PE management, and also to highlight shortcomings in current definitions and prognostic factors. Additionally, we discuss novel therapies currently in preclinical and early clinical trials that may improve outcomes in patients with submassive PE.

## 1. Introduction

Pulmonary embolism (PE) is a common disease with an estimated incidence of approximately 0.5 per 1000 person-years [1,2]. Patients with hemodynamic instability have an exceptionally high mortality rate, approaching 50% [3,4]. In this patient group, the benefits of systemic thrombolysis outweigh the risks of catastrophic bleeding, and overall survival is improved [5,6]. However, the role of systemic thrombolysis in those with hemodynamically stable disease is less clear. While this patient group accounts for approximately 95% of all PE cases, identifying a subgroup for whom thrombolysis provides benefit remains controversial. Attempts to improve outcomes in those with submassive PE, also called intermediate-risk PE, have been disappointing [7]. Challenges include variable definitions of submassive PE, non-standardised criteria of right ventricular dysfunction (RVD), underpowered studies, and lack of all-cause mortality as a primary endpoint. Furthermore, systematic reviews (SRs) have encouraged confusion by reaching different conclusions [8]. In this article, we hope to clarify the current evidence and inform clinical practice. We summarize the definitions, prognostic factors, and management of submassive PE to provide treatment recommendations and discuss novel therapeutic approaches for the treatment of PE. 

## 2. Definitions and Epidemiology

### 2.1. What Is the Definition of Submassive Pulmonary Embolism?

The definition of submassive PE aims to identify a subset of patients with disease severity between massive PE, characterised by hemodynamic instability, and standard-risk PE. RVD is central to this risk-stratification because of its critical role in disease severity; that is, consequent RV failure is the primary cause of death in PE [9,10]. Indeed, the prognostic value of RVD has been affirmed in several studies using both echocardiography [11,12,13,14,15] and CT pulmonary angiogram (CTPA) [16,17,18,19]. Cardiac biomarkers are also used in the definition of submassive PE, with elevated troponins and brain natriuretic peptide (BNP) both associated with increased mortality [20,21,22,23,24,25,26]. 

RVD is often combined with abnormal cardiac biomarkers in defining submassive PE. However, there exists variability in whether one or both criteria are required. Furthermore, which cardiac biomarker should be used also differs between guidelines. This heterogeneity is highlighted in guidelines by the American Heart Association (AHA), American College of Chest Physicians (ACCP), and European Society of Cardiology (ESC), as summarized in Table 1 [27,28,29]. The AHA definition of submassive PE allows for either RVD or elevated troponin levels [27]. Similarly, the ACCP definition of intermediate-risk PE is met if either RVD or abnormal cardiac biomarkers are present, but also includes abnormal BNP in addition to elevated troponins [28]. The ESC adds further complexity by incorporating the Pulmonary Embolism Severity Index (PESI) and simplified PESI (sPESI). These scores consider clinical parameters of PE severity (respiratory rate, hypoxia) as well as patient comorbidities [30,31,32]. Intermediate–high-risk disease requires the presence of both RVD and elevated troponins, with the addition of PESI class III/IV or sPESI ≥ 1. Intermediate–low-risk disease requires PESI class III/IV or sPESI ≥ 1, with or without one of either RVD or elevated troponins [29]. These discrepancies highlight the difficulty of defining an intermediate-risk group of PE patients. Indeed, there is a wide spectrum of clinical severity within all these definitions, and it is likely that those on the severe end stand to benefit more from treatment intensification. The addition of other clinical variables to the current definitions of submassive PE (e.g., respiratory rate) may better identify those in whom thrombolysis is favourable, although this will need to be incorporated into prospective clinical trials. 

### 2.2. What Is the Mortality Associated with Submassive PE?

The mortality rate of submassive PE can vary widely, casting doubt on whether an intermediate-risk group is truly captured. To illustrate, Grifoni et al. observed RVD in 31% of cases from a prospective cohort of 209 hemodynamically stable patients [33]. Those with RVD developed PE-related shock and subsequent mortality in 10% and 5% of cases, respectively. In comparison, none of the patients with normal RV function developed adverse outcomes. Similar mortality rates were observed in the study by Becattini et al., where 906 patients with PE were prospectively stratified into low-, intermediate–low-, intermediate–high-, and high-risk groups according to the 2014 ESC risk stratification model [34]. The 30 day mortality rates were 0.5%, 6.0%, 7.7% and 22%, respectively. The mortality rate was higher in the I-COPER registry, where 1035 patients with normal blood pressure underwent echocardiography within 24 h of acute PE diagnosis [35]. RV hypokinesis was present in 405 (39%) patients, and was associated with a 30 day mortality rate of 16.3%, compared to 9.4% in those without RV hypokinesis. By comparison, these results are drastically different to the more contemporary PEITHO study [36]. Patients with submassive PE, defined by both RVD and elevated troponins, had a mortality rate of only 1.2%. It is unclear why the mortality rate was much lower than previous prospective studies [33,34]. However, one possible explanation is the early intervention and favourable outcomes of patients who suffered hemodynamic deterioration in the anticoagulation arm. This occurred in 25 patients (5%), nearly all of whom received rescue thrombolysis, with only one patient suffering 30 day mortality [36]. 

## 3. Prognostic Markers: Right Ventricular Dysfunction and Cardiac Injury

### 3.1. What Is the Prognostic Significance of Right Ventricular Dysfunction? 

Echocardiographic assessment commonly includes the RV/LV ratio and the tricuspid annular plane systolic excursion (TAPSE) [11,12,13,14]. In a study of 411 patients with hemodynamically stable PE, the hazard ratio (HR) for 30 day PE-related mortality or rescue thrombolysis was increased with both elevated RV/LV ratio (HR 7.3, 95% CI 2.0–27.3, *p* = 0.003) and TAPSE (HR 27.9, 95% CI 6.2–124, *p* < 0.0001) [11]. The positive (PPV) and negative predictive value (NPV) for the same 30 day outcome was 13.2% and 97% with an RV/LV ratio > 1, and 20.9% and 99% with a TAPSE of ≤ 15 mm, respectively. In a larger prospective series, the prognostic significance of TAPSE was evaluated in 782 patients with hemodynamically stable PE [15]. Patients with TAPSE ≤ 1.6 cm were more likely to die from PE at 30 days (HR 2.5, 95% CI 1.3–15.3, *p* = 0.02). 

CTPA can also assess RV dysfunction and is more readily available than echocardiography. The RV/LV ratio measured on transverse sections is the most predictive parameter [16,17,18,19]. In a cohort of 411 patients with hemodynamically stable PE, an RV/LV ratio ≥ 0.9 was associated with increased death or clinical deterioration (HR 3.8, 95% CI 1.3–10.9, *p* = 0.007) [19]. The PPV and NPV for the same outcome was 9% and 97%, respectively. A large meta-analysis evaluated the predictive value of several CTPA parameters [16]. All-cause mortality was increased with an RV/LV ratio ≥ 1.0 (OR 2.5, 95% CI 1.8–3.4, *p* < 0.0001), bowing of the interventricular septum (OR 1.7, 95% CI 1.2–2.4, *p* = 0.0027), and contrast reflux into the inferior vena cava (OR 2.2, 95% CI, 1.5–3.2, *p* < 00001). The predictive value of these parameters applies to unselected PE patients and may not prognosticate those with very low-risk disease. In a study of 779 patients with a sPESI score of zero, an RV/LV ≥ 0.9 or ≥ 1.0 was not associated with worse outcomes [18]. 

### 3.2. What Is the Prognostic Value of Cardiac Biomarkers? 

Cardiac biomarkers include tests of myocardial injury (troponins) and stretch (BNP and N-terminal [NT]-proBNP). Both are associated with increased mortality in normotensive patients [20,21,22]. In a large meta-analysis of 1985 patients with unselected PE, elevated troponins were associated with increased short-term death (OR 5.24; 95% CI 3.28–8.38) [23]. The predictive value was preserved in the subgroup of 915 hemodynamically stable patients (OR 5.9, 95% CI 2.68–12.95). An updated systematic review included 7303 normotensive patients and demonstrated a similar increase in all-cause mortality with abnormal troponins (OR 4.80, 95% CI 3.25–7.08) [24]. Subgroup meta-analysis of studies that used high-sensitive cardiac troponin T (hsTnT) also showed increased death (OR 3.80, 95% CI 2.74–5.27) [24]. In a prospective study of 156 normotensive PE patients, hsTnT levels ≥ 14 pg/mL had a PPV and NPV for adverse 30 day outcomes of 8% and 100%, respectively [25]. A normal hsTnT can therefore help identify patients at low risk of adverse outcomes. Baseline hsTnT levels are higher in older patients, and an age-adjusted cut-off of ≥45 pg/mL has similar predictive value in those aged over 75 years [37]. 

BNP and NT-proBNP are similarly prognostic. In another meta-analysis, short-term mortality was increased in unselected PE patients with either elevated BNP (OR 6.5, 95% CI 2.0–21) or NT-proBNP (OR 8.7, 95% CI 2.8–27) [26]. Similar to troponins, the NPV is nearly 100% and a value less than the cut-off (generally 500 pg/mL for NT-proBNP and 90 pg/mL for BNP) can be used to identify low-risk patients safe for outpatient treatment [20,21]. 

## 4. Treatment of Submassive PE

### 4.1. What Is the General Approach to Treatment? 

Anticoagulation is the mainstay of treatment for patients with PE with reperfusion therapies generally reserved for patients with massive PE. These include systemic thrombolysis, catheter-directed thrombolysis (CDT), mechanical catheter-based techniques, and surgical embolectomy. The precise role of these therapies in patients with submassive PE remains unclear and current guidelines do not support routine use due to a lack of clear clinical benefit [27,28,29]. The evidence-base and the outstanding questions associated with reperfusion in this intermediate group are addressed in the following sections. 

### 4.2. What Is the Optimal Anticoagulation in Submassive PE? 

The initial choice of anticoagulation in patients with submassive PE is generally between low-molecular weight heparin (LMWH) and a direct oral anticoagulant (DOAC). These are preferred over unfractionated heparin (UFH) due to lower rates of bleeding and heparin-induced thrombocytopenia [38,39]. The use of DOACs as an initial PE treatment is supported by the rapid onset of action [40] and noninferiority when compared to enoxaparin followed by warfarin in phase III trials [41,42]. However, the number of patients with submassive PE in these trials is unknown, as RVD was not routinely measured. Additionally, the majority of patients in both rivaroxaban (EINSTEIN) [42] and apixaban (AMPLIFY) [41] studies received LMWH prior to DOAC randomization. This accounts for the early risk of haemodynamic deterioration in the initial 24–48 h, occurring in approximately 5% of patients with submassive PE [36]. In this group of patients, it remains appropriate to initiate treatment with LMWH for 1–2 days before switching to a DOAC. 

### 4.3. Systemic Thrombolysis

#### 4.3.1. What Is the Rationale for Systemic Thrombolysis? 

Systemic thrombolysis causes rapid thrombus resolution, thereby improving pulmonary pressure, RVD, and haemodynamics faster than anticoagulation alone [43,44]. While patients with massive PE have improved mortality rates [5,6,45], its benefit in patients with submassive PE has not been clearly demonstrated [36,43,46,47]. Nonetheless, the role of thrombolysis in submassive PE remains controversial, since thrombolysis does reduce hemodynamic deterioration and has been linked to a reduction in long-term complications such as chronic thromboembolic pulmonary hypertension (CTEPH) [36]. As such, there remains a significant amount of interest in defining the role of intervention in these patients. Furthermore, the development of interventional radiology techniques, such as CDT, and the renewed interest in half-dose thrombolysis have raised the prospect of affording patients the benefits of thrombolysis with reduced bleeding. 

#### 4.3.2. What Is the Evidence of Systemic Thrombolysis in Submassive PE? 

The PEITHO trial is the largest randomised controlled trial (RCT) to date and constitutes a major landmark in the field [36]. It randomised 1005 patients with intermediate-risk PE (requiring both RVD and elevated troponins) to receive tenecteplase or placebo, in combination with UFH [36]. Thrombolysis halved the number of patients meeting the primary endpoint of death or hemodynamic decompensation within 7 days (OR 0.44, 95% CI 0.23–0.87, *p* = 0.02). However, it also increased major extracranial (OR 5.55, 95% CI 2.3–13.39, *p* < 0.001) and intracranial bleeding (2.0% with tenecteplase vs. 0.2% with placebo). Furthermore, the positive primary outcome was driven by a reduction in hemodynamic collapse (OR 0.30, 95% CI 0.14–0.68, *p* = 0.002) rather than death (OR 0.65, 95% CI 0.23–1.85, *p* = 0.42). All-cause mortality at 30 days was similar (2.4% with tenecteplase vs. 3.2% with placebo, OR 0.73, 95% CI 0.34–1.57, *p* = 0.42). 

Other studies have been limited by small numbers and the need for composite endpoints. The Tenecteplase Italian Pulmonary Embolism Study (TIPES) enrolled 51 patients with normal blood pressure and echocardiographic RVD [43]. While thrombolysis reduced the RV/LV ratio at 24 h compared to placebo (*p* = 0.04), the study was too underpowered to measure clinical outcomes. The Management Strategies and Prognosis of Pulmonary Embolism (MAPPET-3) trial was larger and randomised 256 submassive PE patients to receive thrombolysis plus heparin or heparin alone [46]. Thrombolysis reduced the composite endpoint of in-hospital death or clinical deterioration compared to heparin alone (11.0% vs. 24.6%, *p* = 0.006). This was driven by a reduction in clinical deterioration (10.2% vs. 24.6%, *p* = 0.004) with no significant difference in mortality (3.4% vs. 2.2%, *p* = 0.71). The North American Tenecteplase or Placebo: Cardiopulmonary Outcomes at Three Months (TOPCOAT) trial also investigated thrombolysis in patients with submassive PE [47]. There were large differences in the study design as compared to the contemporaneous European PEITHO study. The definition of submassive PE was broader and allowed for either RVD or abnormal cardiac biomarkers (troponins or BNP/NT-proBNP), and echocardiography to assess for RVD was performed in only 65% of patients. The trial was terminated early due to relocation of the primary investigator, and patient numbers were small, with 40 and 43 patients randomised to tenecteplase and placebo, respectively. The 5 day primary outcome was a composite of death, circulatory shock, intubation, or major bleeding, and occurred in one patient treated with thrombolysis compared to three patients treated with heparin alone. 

#### 4.3.3. Why Are Conclusions from Meta-Analyses Different? 

A large number of SRs have assessed the benefits and harms of thrombolytic therapy in PE. Conclusions have been discordant and are a source of confusion. The review by Riva et al. explored these disagreements in twelve SRs published after the initial results of the PEITHO trial [8]. These all demonstrated a concordant reduction in all-cause mortality with thrombolysis. However, only three were statistically significant. In contrast, the results of thrombolysis on major bleeding were discordant: nine SRs demonstrated an increase (five significant) while two demonstrated a decrease (both nonsignificant). The subsequent conclusions regarding the net clinical value of thrombolysis therefore varied between SRs, ranging from strongly positive to strongly negative. 

Surprisingly, meta-analyses that included PEITHO still reached different conclusions [48,49]. For example, a SR of sixteen RCTs (*n* = 2115) by Chatterjee et al. concluded that thrombolysis decreased all-cause mortality in submassive PE [48]. Thrombolytic therapy was associated with lower all-cause mortality (2.17% vs. 3.89%, OR 0.53, 95% CI 0.32–0.88) and a greater risk of major bleeding (9.24% vs. 3.42%, OR 2.73, 95% CI 1.91–3.91). The reduction in all-cause mortality was maintained in the prespecified analysis of eight trials (*n* = 1775) that only enrolled patients with submassive PE (1.39% vs. 2.92%, OR 0.48, 95% CI 0.25–0.92). However, the SR by Nakamura et al. reached a different conclusion [49]. The meta-analysis consisted of six RCTs (*n* = 1510) that compared thrombolysis with heparin in submassive PE. There was no significant difference in all-cause mortality (2.3% vs. 3.7%, RR 0.72, 95% CI 0.39–1.31) or major bleeding (6.6% vs. 1.9%, RR 2.07, 95% CI 0.58–7.35) between the two groups. The authors concluded that thrombolysis did not reduce the risk of mortality in submassive PE. One explanation for the discordant results is the inclusion of different primary studies. Chatterjee et al. included eight additional RCTs in the submassive PE meta-analysis that were ineligible in the Nakamura et al. review: MOPETT and ULTIMA [50,51]. The two SRs otherwise shared the same set of primary studies. The inclusion of MOPETT was unusual, because this trial enrolled patients with ‘moderate’ PE, defined by clot burden rather than RVD or positive cardiac biomarkers. Furthermore, it did not evaluate short-term mortality, and deaths were from 28 month follow-up. The inclusion of the ULTIMA study was also unusual, in that it investigated CDT rather than systemic thrombolysis [51]. 

#### 4.3.4. When Would We Consider Thrombolysis in Submassive Pulmonary Embolism?

The current literature does not support routine systemic thrombolysis in patients with submassive PE. However, several situations may warrant careful deliberation of the risks and benefits. In the case of progressive hemodynamic deterioration, ‘rescue thrombolysis’ is often performed as salvage therapy. Thrombolysis can also be considered in normotensive patients, who deteriorate with respiratory failure despite anticoagulation. Indeed, a PEITHO subgroup analysis demonstrated a significant reduction in the primary outcome with thrombolysis in those with a respiratory rate > 24 [36]. This was not observed in other clinical subgroups such as those stratified by heart rate (≤100 vs. >100 bpm) or symptom onset (≤24 vs. >24 h). The decision to thrombolyse should be weighed against the individual risk of bleeding. In the PEITHO study, major bleeding and intracranial haemorrhage occurred in 11.5% and 2% of patients treated with thrombolysis, respectively. The risk of major extracranial haemorrhage from thrombolysis was higher in those aged over 75 years (OR 20.38, 95% CI 2.69–154.53) compared to younger patients (OR 2.8, 95% CI 1.00–7.86), although the difference was not significant (*p* = 0.09). The effect of age and frequency of major bleeding was also examined in the SR by Chatterjee et al. [48]. In patients aged 65 years and over, thrombolysis was associated with increased major bleeding compared to anticoagulation only (12.93% vs. 4.10%, OR 3.10, 95% CI 2.10–4.56). In comparison, there was no difference between the two groups in those aged less than 65 (2.84% vs. 2.27%, OR 1.25, 95% CI 0.50–3.14). Another important clinical question regarding thrombolysis arises in the context of PE presenting with syncope. Indeed, the presence of syncope has been associated with an increased prevalence of hemodynamic instability and RV dysfunction [52]. However, importantly, the presence of syncope as a presenting feature does not appear to have any significant adverse prognostic significance in patients who are normotensive [52]. Therefore, the current evidence does not support thrombolysis in patients with PE who present with syncope without any evidence of hemodynamic compromise.

### 4.4. Catheter-Based Therapy

#### 4.4.1. What Is the Evidence for Catheter-Directed Thrombolysis? 

Catheter-based therapy has an emerging role in PE management, and includes both CDT and mechanical fragmentation. CDT involves positioning catheters directly into thrombi and delivering local thrombolysis. It can be combined with high-frequency ultrasound waves (US-CDT) which alter the structure of polymerized fibrin, thus enhancing the binding and penetration of tissue plasminogen activator (tPA) into the fibrin rich thrombus [53,54,55]. 

Small trials have demonstrated reduced pulmonary pressures and RVD with US-CDT/CDT [51,56,57,58,59]. The ULTIMA trial compared US-CDT with heparin to heparin alone in 59 hemodynamically stable PE patients with an RV/LV ratio ≥ 1 [51]. The mean RV/LV ratio was significantly reduced at 24 h compared to baseline in the US-CDT group (1.28 ± 0.19 to 0.99 ± 0.17, *p* < 0.001) in contrast to the heparin only group (1.20 ± 0.14 and 1.17 ± 0.20, *p* = 0.31). The SEATTLE II trial was a prospective, single-arm study of 150 massive and submassive PE patients that evaluated US-CDT using the EkoSonic Endovascular System (EKOS, Bothell, Washington) in addition to standard anticoagulation [56]. There was a reduction in the mean RV/LV ratio (1.55 vs. 1.13, *p* < 0.0001), mean PASP (51.4 mm Hg vs. 36.9 mm Hg, *p* < 0.0001), and modified Miller Index score (22.5 vs. 15.8, *p* < 0.0001) within 48 h compared to baseline. Major bleeding occurred in 10% of patients. The OPTALYSE PE trial evaluated different US-CDT regimens that varied in alteplase dose and duration [57]. Submassive PE patients (*n* = 101) were randomised into one of four groups (2 h × 2 mg/h/catheter, 4 h × 1 mg/h/catheter, 6 h × 1 mg/h/catheter, and 6 h × 2 mg/h/catheter). RV/LV ratios and modified Miller scores were significantly reduced in all groups at 48 h. Major bleeding occurred in four patients (4%), two of whom were in the highest dose group. The PERFECT registry prospectively enrolled 101 PE patients that received CDT (64%) or US-CDT (36%) [58]. Clinical success (defined as hemodynamic stabilization, improved pulmonary hypertension or RVD, and survival to hospital discharge) occurred in 85.7% and 97.3% of patients with massive and submassive PE, respectively. Subgroup analyses comparing CDT to US-CDT did not reveal differences in PASP change (*p* = 0.900). While these trials demonstrate the efficacy of CDT in reducing RVD and anatomical clot burden, data regarding clinical endpoints are still lacking. Additionally, the rate of major bleeding was high in some studies (SEATLE II), despite a dose reduction to approximately one fifth of the systemic dose. These events include access-site haematomas as well as spontaneous muscular and intracranial haemorrhages. Dose reductions down to 4 mg, as in the OPTALYSE trial, or even lower may widen the safety profile and favourability between CDT and systemic thrombolysis. However, high-quality RCTs are needed. 

#### 4.4.2. What about Mechanical Catheter-Based Techniques? 

Thrombi can be mechanically disrupted using standard pigtail or angioplasty catheters. However, this technique has become less favourable due to the high-risk (~25%) of embolization [60]. Newer techniques are now available, although the evidence is limited to a few small studies. Rheolytic therapy, performed with the AngioJet device (Boston Scientific), was assessed in 15 patients with massive or submassive PE [61]. This technique uses pressurized saline to macerate the thrombus, which is then aspirated. Resolution of symptoms and improved RVD occurred in all patients. Complications were high (20%) and included two patients with acute tubular necrosis (possible AngioJet-mediated haemoglobinuria) and one patient with an intraoperative cardiac arrest (possible distal embolisation or AngioJet-mediated bradyarrhythmia). Rotational embolectomy, using the Aspirex aspirating spiral catheter (Straub Medical), was performed in 11 of 18 patients with massive PE who did not improve after initial thrombus fragmentation using a routine pigtail catheter [62]. An improvement in haemodynamics without major complications occurred in 16 patients (88.8%). The FLARE study prospectively evaluated thrombus aspiration using the FlowTriever System (Inari Medical) in 106 patients with submassive PE [63]. The RV/LV ratio was significantly reduced at 48 h compared to baseline (*p* < 0.0001). Major adverse events occurred in four (3.8%) patients. Major bleeding and intraprocedural pulmonary haemorrhage occurred in one patient each. Overall, there is early evidence on the benefits of mechanical catheter-based techniques as measured by echocardiographic and CTPA endpoints. Similarly to CDT, further studies incorporating meaningful endpoints and stronger safety data will help to clarify the role of these techniques in PE management. 

## 5. Long-Term Complications after Submassive PE

### 5.1. What Is the Post-PE Syndrome?

Post-PE syndrome refers to a spectrum of long-term complications that include persistent functional limitations, decreased quality of life (QoL), and cardiopulmonary dysfunction [64]. It encompasses mild dyspnoea to the severe CTEPH. The pathophysiology relates to residual pulmonary thrombus and small vessel remodelling, causing arteriopathy and increased vascular resistance [65]. More recently, the role of residual thrombus and RVD in the mechanism of post-PE dyspnoea has been questioned [66,67]. The ELOPE study prospectively evaluated functional limitations in 100 patients with unselected PE [66]. Cardiopulmonary exercise tests demonstrated that 46.5% of patients had a VO_2_ peak of <80% at 1 year follow-up. These patients had worse PE-specific QoL scores, dyspnoea scores, and 6 min walk distance (6MWD) compared to those with a VO_2_ > 80%. Interestingly, there was no difference between the two VO_2_ groups with regards to residual thrombus (CTPA or perfusion scan), pulmonary function test, or echocardiographic RVD. 

### 5.2. How Common Are Chronic Complications after Submassive PE?

The incidence of post-PE syndrome after submassive PE was demonstrated in the PEITHO long-term follow-up study [68]. A total of 709 patients from the original trial underwent clinical and echocardiographic evaluation at 24 months or later after randomization. Persistent clinical symptoms occurred in 36% and 30.1% of patients in the thrombolysis and placebo arm, respectively. While the majority of these patients had mild exertional dyspnoea, a significant proportion still had NYHA class III or IV dyspnoea (12.0% with thrombolysis vs. 10.9% with placebo). RVD was present in more than 35% of patients in each group. Notably, the association between RVD and the presence of symptoms was not reported. The risk of pulmonary hypertension on echocardiogram (determined by ESC criteria) was low in 60% of patients and intermediate in 25%. The diagnostic work-up for CTEPH was performed according to standard medical care and occurred in only 2.6% of patients overall. 

The high rate of persistent RVD has also been reported in other studies [69,70]. Levinson et al. demonstrated RVD in 17% of PE patients at 6 month follow-up [70]. While lower than that observed in PEITHO, this study included unselected PE patients with only 50% having submassive disease. This is an important distinction as chronic RVD is more common in those with submassive PE. These results are consistent with a retrospective study of 508 patients from our own centre [69]. Normotensive patients were stratified into intermediate-risk (*n* = 126) and standard-risk (*n* = 382) by the presence or absence of RVD, respectively. RVD was more common in the intermediate risk group at a median follow-up of 10.8 months (44% vs. 18%, *p* = 0.04). CTEPH was suspected in only three patients (based on echocardiography), none of whom underwent confirmatory right heart catheterisation due to medical comorbidities. 

### 5.3. Does Thrombolysis Reduce the Chronic Complications Post Pulmonary Embolism?

The PEITHO long-term follow-up study provides the strongest evidence on thrombolysis and the prevention of chronic cardiopulmonary complications [68]. Thrombolysis did not reduce the high incidence of post-PE syndrome, and resulted in a similar proportion of patients with persistent symptoms (36% vs. 55%, *p* = 0.23), RVD (44.1% vs. 36.6%, *p* = 0.20), and definitive CTEPH (2.1% vs. 3.2%, *p* = 0.79) compared to placebo. 

However, smaller studies have reported long-term benefits of thrombolysis in preventing chronic complications [50]. The MOPETT trial evaluated low-dose thrombolysis compared to anticoagulation alone in 121 patients with ‘moderate PE’, defined as CTPA involvement of >70% thrombi in ≥2 lobar or left or right main pulmonary arteries, with at least two clinical signs or symptoms of acute PE [50]. The primary endpoint was pulmonary hypertension at 28 months, defined as a pulmonary artery systolic pressure [PASP] ≥ 40 mm Hg on echocardiography. This occurred in an extraordinarily high 57% of patients in the anticoagulation group compared to 16% in the low-dose thrombolysis arm. Several limitations are notable. The results cannot be applied to patients with submassive PE, since the trial enrolled ‘moderate’ cases using criteria based on thrombus load which is not representative of contemporary risk stratification models. Indeed, the degree of clot burden does not reliably predict increased mortality [16]. Additionally, the rate of pulmonary hypertension was drastically higher compared to other studies. By contrast, only 25% of patients in the anticoagulation group had a PASP > 35 mm Hg in the PEITHO long-term study [68]. In our study, only two (1.6%) patients with submassive PE had an echocardiogram suggestive of pulmonary hypertension [69]. 

The TOPCOAT trial also concluded positively, suggesting improved functional outcomes with thrombolysis in patients with submassive PE [47]. The primary outcome at 90 day follow-up was a composite of recurrent PE, poor functional capacity, and poor physical health-related QoL. Poor functional capacity required both echocardiographic RVD and exercise intolerance, the latter defined by a 6MWT distance < 330 m or NYHA class ≥ III. The primary 90 day outcome occurred in five (12.5%) and thirteen (30%) patients in the thrombolysis and placebo group, respectively. The authors combined this with the primary 5 day outcome to achieve a statistically significant reduction in adverse events (six [15%] thrombolysis vs. sixteen [37%] placebo patients, *p* = 0.027). However, there was no difference between thrombolysis and placebo in the individual components of the 90 day composite outcome, and patients that received thrombolysis had a similar proportion of NYHA class ≥ 3 (*p* = 0.051), RVD (*p* = 0.64), and 6MWT distance < 330 m (*p* = 0.19) compared to placebo. The conclusion regarding favourable outcomes with thrombolysis is questionable with small patient numbers, convoluted composite endpoints, and the combination of early outcomes. 

## 6. Future Directions

### 6.1. What about Reducing the Dose of Thrombolysis? 

As discussed, the major benefit of thrombolysis in the context of submassive PE is a reduction in the rate of hemodynamic deterioration. However, this is offset by the risk of major bleeding [36]. Consequently, there remains significant interest in whether lowering the dose of thrombolysis may widen the therapeutic window by affording the same benefit without a prohibitive increase in bleeding [50,71]. 

Few studies have evaluated half-dose thrombolysis in submassive PE [50,71,72]. A prospective Chinese study compared half- and full-dose alteplase in 118 patients with massive and submassive PE [72]. RVD, perfusion defects, and anatomical obstruction were similarly improved in both groups. There was a trend for increased overall bleeding (32% vs. 17%, *p* = 0.054) and major bleeding (10% vs. 3%, *p* = 0.288) in the full-dose group compared to half-dose. However, a significant reduction in overall bleeding was observed with half-dose in patients with body weight < 65 kg (14.8% vs. 41.2%, *p* = 0.049). In contrast, a recent large retrospective study has cast doubt on the efficacy of half-dose thrombolysis [73]. It included 3768 PE patients admitted to the intensive care unit and treated initially with either half- (18.6%) or full-dose alteplase (81.4%). Propensity matching was performed to eliminate differences in disease severity and comorbidities. Treatment escalation occurred more frequently in the half-dose thrombolysis group (54% vs. 41%, *p* < 0.01), driven by a higher occurrence of additional thrombolysis and catheter directed mechanical fragmentation. Further, there was no significant difference in the rates of intracranial haemorrhage (0.5% vs. 0.4%, *p* = 0.67) which occurred at a similar frequency to previous studies of thrombolysis in PE [36]. Limitations include the observational design of the study, thus the decision for a particular thrombolytic dose, as well as the fact that the reason for treatment escalation could not be determined. Overall, the role of half-dose thrombolysis remains unclear with respect to both efficacy and safety. These questions are to be addressed in the upcoming PEITHO-3 trial [74]. Patients with submassive PE will be randomised to a reduced dose of alteplase (0.6 mg/kg to maximum of 50 mg) or placebo. The study will evaluate short-term efficacy and safety at day 30, as well as the effect on long-term outcomes such as functional impairment, residual RVD, and CTEPH. 

### 6.2. What Are the Novel Antithrombotic Strategies for Submassive PE? 

Novel antithrombotic therapies include new anticoagulants, agents that enhance fibrinolysis, and targeted thrombolysis [75]. DOACs are safer when compared to warfarin, but they still carry an approximate 3% annual rate of major bleeding [76]. Attempts to develop the ‘holy grail’ of anticoagulation therapy, a drug that does not cause bleeding, has led to the rational targeting of factor XI (FXI) and XII (FXII). These factors are essential for thrombus propagation but have little role in haemostasis. Patients with congenital FXI or FXII deficiency have a mild bleeding diathesis or no bleeding at all [77,78]. Mice deficient in FXI or FXII demonstrate attenuated thrombosis after venous flow restriction in the inferior vena cava, as well as similar bleeding after tail vein amputation compared to wild-type mice [79,80]. 

These observations have paved the way for the development of therapeutic strategies to target FXI or FXII using antisense oligonucleotides (ASOs) [81], small interfering RNAs, monoclonal antibodies [82,83], and small molecule inhibitors [84] (Table 2). The role of targeting FXI has been evaluated in clinical trials with two phase II studies now published [81,83]. ISIS 416858 is a second-generation FXI-ASO that was prospectively evaluated in 300 patients undergoing unilateral knee arthroplasty [81]. Patients were randomised to two doses of FXI-ASO (200 mg or 300 mg) or enoxaparin 40 mg daily. The 200 mg group was noninferior, while the 300 mg group was superior to the enoxaparin group in reducing the incidence of venous thromboembolism (*p* < 0.001). Bleeding was not significantly different, occurring in 3%, 3%, and 8% in the 200 mg, 300 mg, and enoxaparin groups, respectively. Similarly, a FXIa inhibitory antibody (osocimab) has recently been demonstrated to be noninferior to enoxaparin for VTE prophylaxis after knee arthroplasty [83]. Major or clinically relevant nonmajor bleeding occurred in 4.7%, 4.9%, 2% of patients in the osocimab, enoxaparin, and apixaban arms, respectively. Although yet to enter clinical trials for the treatment or prevention of VTE, FXII-targeted therapeutics have demonstrated efficacy in animal models of thrombosis without impeding haemostasis [85]. While promising, the use of FXI- and FXII-targeted therapeutics for the treatment of an established thrombus, such as acute VTE, remains to be investigated. Moreover, whether the use of these novel antithrombotic approaches are safe in conjunction with thrombolysis is a question unlikely to be addressed in the short-term. 

Patients with submassive PE may benefit from novel therapies that enhance fibrinolysis (Table 3). Given that the benefits of thrombolysis in this patient group are mitigated by the increased bleeding risk, newer strategies to safely harness fibrinolysis are needed. Thrombin activatable fibrinolysis inhibitor (TAFI) and α_2_-antiplasmin are two newly identified targets. TAFI is activated (TAFIa) by the thrombin-thrombomodulin complex after thrombin generation [86]. It negatively regulates fibrinolysis by directly removing C-terminal lysine residues on partially degraded fibrin, preventing plasminogen and tPA binding, and decreasing plasmin [87]. DS-1040 is a small molecular inhibitor of TAFIa that was evaluated in a first-in-human study of 103 healthy subjects [88]. TAFIa activity and clot lysis time were reduced in a dose- and time-dependent manner. Bleeding time remained within the normal range for all doses of DS-1040. A phase I-b study investigating DS-1040 in patients with submassive PE has recently completed recruitment (NCT02923115) [89]. Patients received DS-1040b or placebo in addition to standard of care enoxaparin. The primary outcome was clinically relevant bleeding, with secondary outcomes including thrombus volume reduction and pharmacokinetic assessment. Another negative regulator of fibrinolysis is α_2_-antiplasmin which directly inhibits plasmin. Monoclonal antibodies against α_2_-antiplasmin have markedly amplified clot lysis in animal models [90]. The first-in-human study of TS23, an α_2_-antiplasmin-inactivating antibody, demonstrated a dose-dependent fall in α_2_-antiplasmin and D-dimer levels, with no significant bleeding episodes [91]. However, further studies evaluating α_2_-antiplasmin inhibition in the context of PE are needed. 

Another area of significant interest is the development of targeted thrombolysis strategies [92]. The concept of targeted thrombolysis has emerged as a means to broaden the inherently narrow therapeutic window associated with current thrombolytic agents. The overriding principle is to facilitate the accumulation of lytic drugs at the site of thrombosis, allowing penetration into a fibrin rich clot, and ultimately providing safer and more effective thrombolysis. In this regard, thrombolytic drugs have been conjugated to antibodies that target specific components of a thrombus, such as activated platelets, coagulation factors, or fibrin [92]. Alternatively, thrombolytic drugs have also been packaged into novel drug-delivery systems, such as shear-activated nanoparticles and microbubbles, to try and specifically provide ‘clot’ targeted drug delivery [93,94]. Whilst these approaches have demonstrated efficacy in preclinical animal models [92], they are yet to enter clinical trials and therefore are unlikely to be part of the therapeutic armament in the immediate future. More recently, there has been a significant interest in targeting cellular components as an adjunct to thrombolysis. In this regard, increasing evidence has supported a role for neutrophil extracellular traps (NETs)—weblike structures composed of DNA, histones, and neutrophil granular enzymes—rendering thrombi more resistant to thrombolysis in the setting of ischaemic stroke [95,96]. These findings, coupled with in vitro data demonstrating that DNAse treatment to disassemble NETs can augment tPA mediated fibrinolysis, have led to enthusiasm for such adjunctive therapies and their potential to enhance the efficacy of thrombolysis. However, it remains to be established whether this will be effective in the clinical arena, particularly in the context of submassive PE.

## 7. Summary

Submassive PE remains a challenging clinical problem for the hospital physician. The evidence to date does not support the use of routine thrombolysis in this patient group. The benefits of reduced hemodynamic deterioration are outweighed by increased rates of major bleeding, although it is likely a subgroup of patients with severe submassive PE may still derive net gain. Indeed, current definitions of submassive PE are confusing and may not capture a true ‘intermediate-risk’ group, as evidenced by the low rates of mortality (<5%) seen in PEITHO trial and various meta-analyses. Long-term consequences of PE are beginning to be recognized; with post-PE syndrome entering the literature, future work will be required to fully understand this condition. While the immediate focus largely pertains to defining the role for catheter-based therapies and half-dose thrombolysis, exciting advances have yielded new antithrombotic therapies and novel fibrinolytic approaches. It is hoped that these therapies may eventually translate into improved short and long-term outcomes in patients with submassive PE. 

## Figures and Tables

**Table 1 jcm-10-03383-t001:** Definitions of submassive/intermediate-risk PE demonstrating the variable inclusion of RVD and cardiac biomarkers.

	Hemodynamic Instability	RVD	Elevated Cardiac Biomarkers	PESI Class III-IV or sPESI ≥ 1
AHA				
Low risk	No	No	No	NA
Submassive	No	Yes	No	NA
		No	Troponins	
		Yes	Troponin	
Massive	Yes	NA	NA	NA
ACCP				
Low risk	No	No	No	NA
Intermediate risk	No	Yes	No	NA
		No	Troponins and/or BNP	
		Yes	Troponins and/or BNP	
High-risk	Yes	NA	NA	NA
ESC				
Low risk	No	No	No	0
Intermediate-low risk	No	No	No	Yes
		No	Troponins	
		Yes	No	
Intermediate-high risk	No	Yes	Troponins	Yes
High-risk	Yes	NA	NA	Yes

Abbreviations: AHA = American Heart Association; ACCP = American College of Chest Physicians; ESC = European Society of Cardiology; RV = right ventricular; PESI = pulmonary embolism severity index; sPESI = simplified PESI. The colour: risk stratification.

**Table 2 jcm-10-03383-t002:** Novel anticoagulant therapies and clinical development.

Drug Type	Drug	Trial Phase	Trial Status	Trial Description	Results	References
**Factor XI targets**						
ASO	ISIS 416858	Phase 2	Published	ISIS 416858 vs enoxaparin for thromboprophylaxis in patients undergoing TKA	See text	[81]
		Phase 2	Completed	ISIS 416858 in patients with ESRF on haemodialysis (EMERALD)	NA	
Monoclonal antibody	Osocimab	Phase 2	Published	Osocimab vs enoxaparin or apixaban for thromboprophylaxis in patients undergoing TKA (FOXTROT)	See text	[83]
	MAA868	Phase 0	Published	First-in-human single ascending dose study in healthy subjects	Dose-dependent prolongation of APTT and reduction in FXI levels with no adverse bleeding	[82]
		Phase 1	Completed	Dose-range finding study in patients with AF	NA	
	BAY1831865	NA	NA	NA	NA	
	AB023/Xisomab 3G3	Phase 1	Published	First-in-human single ascending dose study in healthy subjects	Dose-dependent duration of limited anticoagulation without increased bleeding	
		Phase 2	Completed	AB023 at the beginning of a regular haemodialysis session in patients with ESRF	NA	
Molecular inhibitor	JNJ-70033093	Phase 1	Published	PK study in patients with mild to moderate hepatic impairment	JNJN-70033093 well-tolerated in healthy participants and those with mild or moderate hepatic impairment	[84]
		Phase 2	Recruiting	Dose-ranging study of BMS-986177 following acute ischaemic stroke or TIA	NA	
		Phase 2	Active, not recruiting	JNJ-70033090 vs enoxaparin for thromboprophylaxis in patients undergoing TKA	NA	
	EP-7041, ONO-5450598	NA	NA	NA	NA	
Aptamers	11.16, 12.7	NA	NA	NA	NA	
**Factor XII targets**						
Monoclonal antibody	9A2, 15H8, AB052 *, CSL312 *, 3F7 *	NA	NA	NA	NA	
Molecular inhibitor	rHA-infestin 4, FXII618, 3-carboxamide coumarins	NA	NA	NA	NA	
Aptamers	R4cXII-1	NA	NA	NA	NA	

Abbreviations: ASO = antisense oligonucleotide; TKA = total knee arthroplasty; PK = pharmacokinetics; AF = atrial fibrillation; ESRF = end stage renal failure; TIA = transient ischaemic attack; NA = not applicable; * Targeting factor XIIa.

**Table 3 jcm-10-03383-t003:** Novel fibrinolytic therapies and clinical development.

Drug Type	Drug	Trial Phase	Trial Status	Trial Description	Results	References
**TAFIa targets**						
Molecular inhibitor	DS-1040	Phase 1	Published	First-in-human single ascending dose study in healthy subjects	See text	[88]
		Phase 1b	Completed	Single ascending dose study when added to standard anticoagulation in patients with acute submassive PE	NA	
		Phase 1b/2	Completed	Single-ascending dose study in acute ischaemic stroke	NA	
	S62798	NA		NA	NA	
Monoclonal antibody	MA-T9H11, MA-RT30D8, MA-TCK11A9, MA-TCK26D6, MA-T12D11, mAbTAFI/TM#16, MA-TCK27A4	NA	None	NA	NA	
**α2-antiplasmin targets**						
Monoclonal antibody	TS23	Phase I	Published abstract	First-in-human ascending dose study (NAIL-IT)	See text	[91]

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
