# Peer review of "Submassive Pulmonary Embolism: Current Perspectives and Future Directions"

_jcm, 2021, doi:10.3390/jcm10153383_

Round 1

Reviewer 1 Report

Dear AUTHORS,

Below, please find my revision of the article entitled Submassive pulmonary embolism: current perspectives and future directions”, by P. Nguyen et al (Manuscript ID jcm-1295863).REVISION

Nguyen et al summarize in this review the definition, epidemiology and management of submassive pulmonary embolism.

Although originality is not its main strength since different other reviews on the subject have been published, the article gives a complete and interesting scientific perspective on an important topic for clinicians and researchers, and the matter seems within the scope of the journal.

The main questions with regards to submassive PE are well described and analyzed.

The methodological approach is very interesting, with very sound clinical questions introducing every topic.

Claims and conclusions are supported by the evidence provided.

Data and evidence are appropriately discussed in the context of the previous literature.

General presentation of the work is clear, and the manuscript is well-written.

No significant concerns arise regarding ethics and COIs.

Overall, this is a high quality manuscript that has implications for clinical practice.

Author Response

We thank the reviewer for their positive and constructive comments.

Reviewer 2 Report

The manuscript is interesting an well written. I have only two suggestions:

  • Please discuss in more details the approach for submassive PE patients with syncope
  • - Please review the mansucript for some minor typing error

Author Response

We thank the reviewer for their overall positive comments and highlighting the important issue of the management of PE presenting with syncope. We have amended the manuscript accordingly to now briefly discuss this important issue. In addition, we have also thoroughly proof read the manuscript to edit any minor typographical errors.